# Inhibition of Cytosolic Phospholipase A2α Induces Apoptosis in Multiple Myeloma Cells

**DOI:** 10.3390/molecules26247447

**Published:** 2021-12-09

**Authors:** Nur Mahammad, Felicity J. Ashcroft, Astrid J. Feuerherm, Samah Elsaadi, Esten N. Vandsemb, Magne Børset, Berit Johansen

**Affiliations:** 1Department of Biology, Norwegian University of Science and Technology (NTNU), 7491 Trondheim, Norway; felicity.ashcroft@ntnu.no (F.J.A.); astrid.j.feuerherm@ntnu.no (A.J.F.); 2Center for Myeloma Research, Department of Clinical and Molecular Medicine, Faculty of Medicine and Health Science, Norwegian University of Science and Technology (NTNU), 7491 Trondheim, Norway; samah.elsaadi@ntnu.no (S.E.); esten.vandsemb@ntnu.no (E.N.V.); magne.borset@ntnu.no (M.B.); 3Department of Immunology and Transfusion Medicine, St. Olav’s University Hospital, 7491 Trondheim, Norway

**Keywords:** cPLA2α, *PLA2G4A*, cPLA2α inhibitor, AVX002, AVX420, multiple myeloma, JJN3, IH1, RPMI8226, INA6, apoptosis

## Abstract

Cytosolic phospholipase A2α (cPLA2α) is the rate-limiting enzyme in releasing arachidonic acid and biosynthesis of its derivative eicosanoids. Thus, the catalytic activity of cPLA2α plays an important role in cellular metabolism in healthy as well as cancer cells. There is mounting evidence suggesting that cPLA2α is an interesting target for cancer treatment; however, it is unclear which cancers are most relevant for further investigation. Here we report the relative expression of cPLA2α in a variety of cancers and cancer cell lines using publicly available datasets. The profiling of a panel of cancer cell lines representing different tissue origins suggests that hematological malignancies are particularly sensitive to the growth inhibitory effect of cPLA2α inhibition. Several hematological cancers and cancer cell lines overexpressed cPLA2α, including multiple myeloma. Multiple myeloma is an incurable hematological cancer of plasma cells in the bone marrow with an emerging requirement of therapeutic approaches. We show here that two cPLA2α inhibitors AVX420 and AVX002, significantly and dose-dependently reduced the viability of multiple myeloma cells and induced apoptosis in vitro. Our findings implicate cPLA2α activity in the survival of multiple myeloma cells and support further studies into cPLA2α as a potential target for treating hematological cancers, including multiple myeloma.

## 1. Introduction

Hematologic malignancies, which are cancers of the blood, bone marrow, and lymph nodes, account for approximately 8% of all cancers [1]. Among them, Multiple Myeloma (MM), characterized by the aggregation of clonal, cancerous plasma cells in the bone marrow, accounts for about 10% of all hematological malignancies [1,2]. It evolves from an asymptomatic premalignant stage termed “monoclonal gammopathy of undetermined significance (MGUS)” and is associated with the overproduction of a monoclonal immunoglobulin or M protein in blood and urine [3]. Multiple myeloma generally affects elderly patients (>65 years) with a median age at diagnosis of 70 years [4]. While newer treatments including thalidomide, lenalidomide (mechanism still under investigation), and proteasome inhibitors (e.g., bortezomib) have been developed during the past 15–20 years, chemotherapy in combination with steroids remains the major treatment for newly diagnosed multiple myeloma. The response rate is high, but all the patients eventually relapse [5,6]. Since multiple myeloma persists as cureless with high relapse frequency, a new therapeutic target could present possibilities for understanding the biology and treatment of the disease.

The phospholipase A2 (PLA2) superfamily of enzymes are present in healthy and cancerous cells. They hydrolyze fatty acids from membrane phospholipids and can provide precursors for the biosynthesis of eicosanoids. Eicosanoids are biologically active lipids with clear roles in various pathological processes such as inflammation and cancer progression [7]. PLA2 enzymes are classified into six main groups: cytosolic and calcium-dependent PLA2 (cPLA2), calcium-independent PLA2 (iPLA2), secreted PLA2 (sPLA2), platelet-activating factor acetyl-hydrolase (PAF-AH), lysosomal PLA2 (lys-PLA2) and adipose PLA2 (Ad PLA2) [8]. cPLA2α is the sole member of the group that shows high selectivity towards phospholipids carrying arachidonic acid (AA) at the *Sn-2* position leading to the release of AA, which serves as the precursor molecule in the biosynthesis of eicosanoids, including prostaglandins (PGs), thromboxanes (TXAs), leukotrienes (LTs), and lipoxins (LXs) [9]. The importance of eicosanoids in several disease settings makes cPLA2α an interesting target for research.

Overexpression of the gene encoding cPLA2α (*PLA2G4A*) has been reported in patient samples from different solid cancers such as breast cancer [10], hepatocellular carcinoma [11], cervical cancer [12], prostate cancer [13], and lung cancer [14], as well as hematological cancers such as acute myeloid leukemia (AML) [15,16] and B-cell lymphocytic leukemia (B-CLL) [17]. Moreover, high gene expression of *PLA2G4A* has been correlated with poor patient outcomes, e.g., relapse, development of metastasis and lower survival, in cancers of lung, liver, and breast, glioblastoma, and multiple myeloma [18,19,20,21,22]. To our knowledge, the role of cPLA2α in multiple myeloma has not however been investigated further.

As an attractive therapeutic target for chronic inflammatory diseases and cancers [23], several molecules have been developed to study or inhibit cPLA2α, and studies have shown that these inhibitors can reduce tumor growth and angiogenesis in solid tumors [24,25,26]. Arachidonyl trifluoromethyl ketone (AACOCF3), for example, was reported to sensitize tumors to radiation therapy via effects on the tumor vasculature [24] and was reported to inhibit the migration and invasion of lung cancer cells in vitro [27]. Another cPLA2 inhibitor, 4-[2-[5-chloro-1-(diphenylmethyl)-2-methyl-1H-indol-3-yl]-ethoxy]benzoic acid (CDIBA), has been used to demonstrate key regulatory roles of cPLA2 and lysophospholipids in brain and lung cancers in vivo [25], and cPLA2α inhibitors (e.g., AVX001, AVX002, AVX235) developed by Avexxin, now Coegin Pharma, were previously shown to inhibit inflammation [28,29,30,31,32,33], tumor progression, and angiogenesis both in vitro and in vivo [34,35]. We previously reported that AVX001 and AVX002 were more potent than AACOCF3 in an in vitro cPLA2α activity assay - where docosahexaenoic acid (22:6, n-3; DHA) was inactive [29]. In the current study, we used AVX002 and AVX420 (Methyl 2-(2-(4-heptyloxy)-phenoxy)-acetyl)thiazole-4-carboxylate) [36,37] to investigate cPLA2α as a potential target for treating the hematological cancer multiple myeloma.

## 2. Results

### 2.1. cPLA2α Is Overexpressed in Cancers and Cancer Cell Lines from Different Tissue Origins

It is known that cPLA2α can be overexpressed in cancer and involved in disease progression [10,11,12,13,14,15,16,17]; however, it is unclear how common this is and whether cancers originating from specific tissues may be more reliant on cPLA2α than others. To get a better overview of whether certain cancers may be more reliant on cPLA2α activity than others, we explored gene expression data from publicly available clinical cancer patient and cell line databases using Genevestigator and the Cancer Cell Line Encyclopedia (CCLE), respectively. Genevestigator is a new generation web-based tool that provides categorized quantitative information about genes and annotations contained in a large microarray database [38]. Analysis of 24,375 patient samples from cancers of 12 different tissue origins showed the highest expression of the gene encoding cPLA2α (*PLA2G4A*) in cancers originating from the respiratory system and skin (Figure 1A). CCLE is a database composed of gene expression, gene copy number, and sequencing data from 947 human cancer cell lines [39]. Analysis of RNA-seq and copy number data using the CCLE showed that the *PLA2G4A* gene was most highly expressed in melanoma and certain leukemia and lymphoma cell lines, including acute myeloid leukemia (AML) and multiple myeloma (MM) (Figure 1B,C).

### 2.2. Hematological Cancer Cells Are Sensitive to cPLA2α Inhibition

A second approach to investigate whether the origin of the cancer was a determinant of the importance of cPLA2α in disease progression, was to determine the effect of cPLA2α inhibition on the viability of a panel of 66 cancer cell lines (Oncolines panel); representing 14 different tissues of origin as shown in Figure 2A. The cPLA2α inhibitors used in this study were AVX002, AVX235, and AVX420. We have previously published the anti-cancer properties of AVX235 in a basal-like breast cancer model [34,35] and reported the anti-inflammatory properties of AVX002 using both in vivo and cellular models [33]. AVX420 ((Methyl 2-(2-(4-heptyloxy)-phenoxy)-acetyl)thiazole-4-carboxylate) is structurally related to AVX235 (Methyl 2-(2-(4-octylphenoxy)-acetyl)thiazole-4-carboxylate) as opposed to AVX002, which is a derivative of omega-3 polyunsaturated fatty acid (ω-3 PUFA) [29,34]. Hematological cancer cell lines were highly sensitive to treatment with all the three cPLA2α inhibitors, and this was the only group for whom the average IC50 was significantly different from the average for the entire panel (Figure 2C–E). The average IC50 values for solid cancers were 19.5 µM, 16.3 µM, and 10.5 µM for AVX420, AVX235, and AVX002, respectively. The average IC50 values were significantly lower in blood cancer cell lines, with IC50 values of 8.5 µM, 11.9 µM, and 7 µM for AVX420, AVX235, and AVX002, respectively (Figure 2B).

### 2.3. cPLA2α Is Overexpressed in Hematological Cancers including Multiple Myeloma

Based on the finding that hematological cancer cell lines were more sensitive to cPLA2α inhibition than solid cancer cell lines, we next explored *PLA2G4A* gene expression in different cancers of blood origin. Analysis of 10,131 hematological cancer patient samples showed the highest gene expression of *PLA2G4A* was in lymphoma, acute myeloid leukemia (AML), and multiple myeloma (MM) (Figure 3A). A separate analysis of the hematological cancer cell lines showed multiple myeloma had the highest *PLA2G4A* copy number (Figure 3B) and that the highest *PLA2G4A* gene expression was in acute myeloid leukemia (AML) lines (Figure 3C).

To compare the gene expression of *PLA2G4A* between cancer patients and healthy individuals, we used the cancer microarray database “Oncomine”, which contains 65 gene expression datasets from over 4700 microarray experiments [40]. We found higher expression of *PLA2G4A* in multiple myeloma patients, as well as patients with the asymptomatic premalignant stage of multiple myeloma known as MGUS, in comparison to healthy individuals. There was approximately a 3-fold increase in *PLA2G4A* gene expression in multiple myeloma and a 2.3-fold increase in MGUS patient samples (Figure 3D,E).

Together these results suggest that the growth of hematological cancers might be more sensitive to cPLA2α inhibition than solid tumors. Overexpression of the *PLA2G4A* gene in multiple myeloma cell lines, and samples from patients with either symptomatic or asymptomatic myeloma, led us further to investigate the role of cPLA2α in multiple myeloma using MM cell lines as models.

Based on RNA sequencing data obtained from Jonathan Keats’ lab (www.keatslab.org, accessed on 8 February 2021) (Figure 3F) and our in-house screening (Figure 3G), four multiple myeloma cell lines (RPMI8226, INA6, IH1, and JJN3) were selected for further investigation. The cell lines RPMI8226 and INA6 had higher expression of the *PLA2G4A* gene than JJN3 and IH1 (Figure 3F,G).

### 2.4. Inhibition of cPLA2α Reduces Cell Viability of Multiple Myeloma Cells

The effect of cPLA2α inhibition on the viability of the four multiple myeloma cell lines was measured using two viability assays (Resazurin assay and Cell Titer Glo assay). AVX002 and AVX420 were used to inhibit cPLA2α; these chemically distinct cPLA2α inhibitors had the highest efficacy (lowest IC50 values) in the Oncolines panel. Both inhibitors dose-dependently reduced the viability of all four cell lines, and comparable IC50 values were observed across both the viability assays (Figure 4).

The sensitivity to AVX420 and AVX002 varied across cell lines (Figure 4C,F). RPMI8226 and INA6 cell lines, representing the high expression of the cPLA2α gene (*PLA2G4A)*, were significantly more sensitive to treatment with AVX002 than AVX420. There was no significant difference in sensitivity between AVX420 and AVX002 in the IH1 and JJN3 cells, although JJN3 cells tended to be more sensitive to AVX420. Collectively, these findings indicate that inhibition of cPLA2α reduces the viability of multiple myeloma cells independent of the gene expression level.

### 2.5. cPLA2α Inhibitors Induce Apoptosis in Caspase-3 Dependent Pathways

Having shown that the cPLA2α inhibitors, AVX420 and AVX002, reduced cell viability in a dose-dependent manner, we next wanted to investigate whether the reduced viability could be explained by apoptosis. Apoptosis was first measured using Annexin V-FITC (fluorescein isothiocyanate) staining. JJN3 cells were treated with cPLA2α inhibitors for 72 h under serum-reduced conditions (4% serum). The percentage of living (Annexin V and propidium iodide negative), early apoptotic (Annexin V positive), and late apoptotic/dead cells (Annexin V and propidium iodide positive) are shown with representative density plots (Figure 5A). Treatment of JJN3 cells with 20 µM of AVX420 or AVX002 significantly increased the numbers of early apoptotic and late apoptotic/dead cells in the population (Figure 5B).

Since both extrinsic and intrinsic pathways of apoptosis converge to a common execution phase involving proteolysis and activation of caspase-3 and/or -7 (caspase-3/7), we next assayed for apoptosis by measuring the activation of caspase-3/7 using a specific substrate. JJN3 cells were treated with AVX420 or AVX002 for 6 h, 18 h, and 48 h. The cells showed significantly increased caspase-3/7 activity after treatment with 20 µM AVX420 or AVX002 for 6 h or 18 h compared to the untreated control. A reduction in the activity of caspase-3/7 after 48 h could indicate the end of the early phase of apoptosis (Figure 5C).

Other early events in apoptosis include the cleavage of poly (ADP-ribose) polymerase (PARP) and caspase-3 [41,42,43]. We analyzed these events using immunoblotting and showed that following 36 h of treatment with 10 µM of either inhibitor; there was an increased level of cleaved PARP and cleaved caspase-3 (Figure 5D). Doxorubicin was used as a positive control in the study, which is a well-established inducer of apoptosis in several other cancer cells [44,45,46].

Together, the results of the Annexin V apoptosis assay, caspase-3/7 activity assay, and immunoblots for cleaved caspase-3 and PARP make it likely that the observed reduction in viability seen in multiple myeloma cells treated with cPLA2α inhibitors is due to induction of apoptosis.

## 3. Discussion

In this study, we showed that cell lines of hematological origin were highly susceptible to the effects of cPLA2α inhibition on cell viability. Multiple myeloma (MM) represented hematological cancer that overexpressed *PLA2G4A*, and we showed that two cPLA2α inhibitors, AVX420 and AVX002, reduced viability in different multiple myeloma cell lines, likely via induction of apoptosis.

Our exploration of public data demonstrated higher expression of *PLA2G4A* in human cancers of specific tissue origins, including the respiratory system, skin, digestive organs, and urinary organs. This is consistent with previous studies in non-small cell lung carcinoma (NSCLC) [47], prostate cancer [13], cholangiocarcinoma (SG231) [48], and colon cancer [49]. We also found *PLA2G4A* to be highly expressed in certain hematological cancers, including acute myeloid leukemia (AML), which is consistent with the previous reports of overexpression of *PLA2G4A* in leukemias [15,16,17], and also in multiple myeloma and lymphoma, which has not previously been reported. We also found higher gene expression of *PLA2G4A* in both MM and MGUS patients than in healthy individuals, supporting MM as an interesting cancer for further investigation.

A growing body of evidence implicates cPLA2α in the development or progression of solid [10,11,12,13,14] as well as hematological cancers [15,16,17] with roles demonstrated in cell proliferation [50,51,52], angiogenesis [24,25,34,53], and metastasis [27,35,54]. Here we show, for the first time, that the cPLA2α inhibitor AVX002, for which only anti-inflammatory properties had previously been described [33], can also affect cell viability, and we report that a novel cPLA2α inhibitor AVX420 had similar effects.

The viability of four cell lines was significantly reduced by cPLA2α inhibitors AVX420 and AVX002, and we investigated whether the reduced viability was due to apoptosis. Structural modification of the plasma membrane with externalization of phosphatidylserine is an early indicator of apoptosis [55,56,57] and was detected in response to both the inhibitors. Since caspase-3/7 plays an essential role in the execution of apoptosis, increased activity of caspase-3/7 can also be used as an indicator of apoptotic cell death [58,59,60]. In our study, caspase-3/7 activity was significantly increased in cPLA2α inhibitor-treated cells at early (6 and 18 h) vs. later (48 h) timepoints. The reduction in caspase-3/7 activity after longer incubation periods that we observed in our study was also reported in a previous study by Terri Sundquist [61]. We also found increased levels of cleaved caspase-3 and cleaved PARP protein in cPLA2α inhibitor-treated cells. These are hallmarks of apoptotic cell death, not occurring in necrosis, and this lends support to the hypothesis that cPLA2α inhibitors can trigger apoptosis in multiple myeloma cells.

Our findings that the AVX002 and AVX420 can reduce the viability of multiple myeloma cells in vitro may be specific to the disease model because AVX235 and AVX002 showed surprisingly low adverse effects in our previous in vivo studies [33,34]. AVX235, which is closely related to AVX420, was tested in a mouse orthotopic xenograft model [34], and AVX002 was tested in both prophylactic and therapeutic collagen-induced arthritis models [33]. In both cases, the cPLA2α inhibitors showed minimal side effects, supporting the potential for using these compounds therapeutically.

cPLA2α inhibition can reduce the cellular production of PGE2 by decreasing the availability of arachidonic acid (AA), reducing the expression of genes encoding cyclooxygenase-2 (COX-2) (*PTGS2*) and cPLA2α (*PLA2G4A*) [30] and inhibiting the activity of the transcription factor nuclear factor kappa B(NFκB) [62]. This has well-known anti-inflammatory effects but can also affect cell survival and proliferation via inhibition of EP4-dependent PI3K/AKT signaling [63] and the transcription factor forkhead box protein O1 (FOXO1) [50]. Based on these studies, we hypothesize that PGE2/EP4/PI3K signaling may be required for the survival of multiple myeloma cells, and thus inhibiting this pathway could result in cell cycle arrest and apoptosis, as summarized in Figure 6.

Based on our findings, we conclude that cPLA2α may have an important role in hematological cancers. Inhibition of cPLA2α in multiple myeloma cells likely causes cell death by apoptosis, supporting its potential as a therapeutic target for the treatment of this disease.

## 4. Materials and Methods

### 4.1. Materials

Cell culture media, recombinant IL-6 (#SRP3096) and dimethyl sulfoxide (DMSO) (#2650) were purchased from Sigma-Aldrich (St. Louis, MO, USA). Antibodies were obtained from Cell Signaling Technology (Danvers, MA, USA) (e.g., cleaved PARP (CST #5625), cleaved caspase-3 (CST #9664) and β-Actin (CST #4970)) and doxorubicin from Cayman Chemicals (Ann Arbor, MI, USA) (#15007). AVX002 (1,1,1-trifluoro-3-(((3Z,6Z,9Z,12Z,15Z)-octadeca-3,6,9,12,15-pentaen-1-yl)thio)propan-2-one) [64] and AVX235 (methyl 2-(2-(4-octylphenoxy)acetyl)thiazole-4-carboxylate) [65] were synthesized by Synthetica AS, Oslo. AVX420 ((Methyl 2-(2-(4-heptyloxy)-phenoxy)-acetyl)thiazole-4-carboxylate) were synthesized in the laboratory of organic chemistry at University of Athens by George Kokotos [36]. AVX002, AVX235 and AVX420 were stored at −80 °C in DMSO.

### 4.2. Maintenance of Multiple Myeloma Cell Lines

The human multiple myeloma cell lines (HMCL) were obtained from St. Olavs University hospital, Trondheim, Norway (IH1), American Type Culture Collection (ATCC) (Manassas, VA, USA) (RPMI8226), a gift from Dr. M Gramatzki, University of Erlangen Nuremberg, Erlangen, Germany (INA6), and Dr. I.M. Franklin, University of Birmingham, UK (JJN3).

All HMCL were maintained in RPMI 1640 supplemented with L-glutamine (0.68 mM) and FBS. All cell lines were cultivated with 10% FBS except RPMI8226 which was cultivated with 20% FBS at 37 °C with 5% CO_2_ in a humidified atmosphere. IH1 and INA6 cells were always supplemented with IL-6 (1 ng/mL). New cells were routinely tested to ensure the absence of *mycoplasma*. Typically, HMCL were maintained in T-75 flasks and split twice per week.

### 4.3. Viability Screening of Oncolines Cancer Cell Panel

Cell viability in response to cPLA2α inhibitors was investigated using the Oncolines cancer cell panel and performed and analyzed according to NTRC’s (Netherlands Translational Research Center) methods described previously (32, 33) except for assaying under reduced serum conditions. In brief, cells were diluted in their ATCC-recommended medium and dispensed in a 384-well plate at a concentration of 200–3200 cells per well in 45 µL medium. Plated cells were incubated in a humidified atmosphere of 5% CO_2_ at 37 °C. After 24 h, 5 µL of the cPLA2α inhibitors with a dose range of 10 nM to 100 µM was added, and plates were incubated for another 72 h. After 72 h, 25 µL of ATPlite 1step™ (PerkinElmer, Waltham, MA, USA) solution was added to each well and subsequently shaken for 2 min. After 10 min of incubation in the dark, the luminescence was recorded on an Envision multimode reader (PerkinElmer, Waltham, MA, USA).

### 4.4. Cell Viability Assays

Cells were harvested by centrifugation at 1500 rpm for 8 min. The supernatant was discarded, and the cells were resuspended in experimental media. The cell counter Countess^TM^ was used to count the number of cells and check the viability by staining with trypan blue (0.4%) (NanoEnTek, Waltham, MA, USA). Then, 10,000 cells were seeded in at least three technical replicates for both CTG (cell titer Glo assay) and resazurin assay in 100 µL and treated as indicated before measurement of cell proliferation. CTG reagent was added according to the manufacturer’s instructions and left to incubate for 10 min, covered with aluminum foil to protect the reaction mix from light. Luminescence was determined using a Victor 1420 multilabel counter (PerkinElmer Inc. Waltham, MA, USA).

Resazurin (RnD Systems, Abingdon, UK) was added according to the manufacturer’s instructions and left to incubate for 2 h at 37 °C with 5% CO_2_ in a humidified atmosphere. Fluorescence was read at 544 nm excitation and 590 nm emission wavelengths using the Cytation 5 cell imaging multimode reader (Biotek Instruments, Winooski, VT, USA).

### 4.5. Annexin V-FITC and Propidium Iodide Apoptosis Assay

Annexin-V-fluorescein isothiocyanate (FITC) kit (Tau Technologies, Albuquerque, NM, USA) was used to determine apoptosis. Cells were harvested by centrifugation at 1500 rpm for 8 min. The supernatant was discarded, and the cells were resuspended in experimental media. The cell counter Countess^TM^ was used to count the number of cells and check viability by staining with trypan blue (0.4%) (NanoEnTek, Waltham, MA, USA). For Annexin V-FITC and propidium iodide staining, 100,000 cells were seeded in 24-well flat-bottom plates. Cells were treated on the same day, and stimulated cells were incubated at 5% CO_2_, 37 °C for 72 h before incubating them with Annexin V-FITC (0.2 μg/mL in 1X Annexin binding buffer) for 1 h on ice. Propidium iodide (1.4 μg/mL) was added 5 min prior to data acquisition using an LSRII flow cytometer (BD Biosciences, San Jose, CA, USA). FlowJo Software v10.1 (Ashland, OR, USA) was used to analyze the data.

### 4.6. Caspase-3/7 Activity Assay

Caspase activity was measured using a Caspase-3/7 Glo assay kit (Promega Corporation, Madison, WI, USA). HMCL cells were seeded in 96-well white plates at a density of 10,000 cells per well. The cells were treated with inhibitors for 6 h, 18 h, and 48 h, then 50 µL caspase-3/7 reagent was added to each well and incubated for 1 h at room temperature. The luminescence reading was then measured using the victor 1420 multilabel counter (Perkin Elmer Inc., Waltham, MA, USA).

### 4.7. Western Blot

To prepare the samples, the cells were washed with ice-cold phosphate-buffered saline (PBS), and lysis buffer (1% of IGEPAL®CA-630 (Sigma-Aldrich, St. Louis, MO, USA), 150 mM NaCl, 50 mM Tris–HCl pH 7.5, 10% glycerol, 50 mM NaF, 1 mM Na_3_VO_4_, and a protease-phosphatase inhibitor mixture (Complete mini tablets; Roche, Basel, Switzerland)) was added for 30 min on ice. Cell debris was discarded by centrifugation at 12,000× *g*, 4 °C for 10 min. Protein concentrations were measured using Quick Start™ Bradford 1X Dye Reagent (Bio-Rad, CA, USA) and iMark™ Microplate Reader (Bio-Rad, CA, USA).

The samples were diluted in lysis buffer to obtain equal protein concentration. Then, they were mixed with lithium dodecyl sulfate sample buffer (Invitrogen, Waltham, MA, USA) with 10 mM dithiothreitol (DTT), heated for 10 min at 70 °C, and separated on 4–12% Bis-Tris gels with MES running buffer (Invitrogen, Waltham, MA, USA). The gel was then transferred to iBlot gel transfer stacks. Proteins were transferred to the nitrocellulose membrane using the iBlot Dry Blotting System (Invitrogen, Waltham, MA, USA). The finished blot was soaked with 5% BSA in Tris-buffered saline with 0.05% Tween-20 and incubated with antibodies diluted to 1:1000 against indicated proteins on a shaker overnight at 4 °C. Super Signal (West Femto Maximum Sensitivity Substrate) (Thermo Fisher Scientific, Rockford, IL, USA) substrate was added to the horseradish peroxidase-conjugated antibodies (Dako Cytomation, Copenhagen, Denmark) stained blot for 3 min before the luminescence signal was detected using odyssey Fc imager (LI-COR Biosciences, Lincoln, NE, USA) and processed using Image Studio software (LI-COR Biosciences, Lincoln, NE, USA).

### 4.8. Statistical Analysis

All analysis in cellular profiling was performed in the statistical software package R. For statistical analysis and generating figures for viability and apoptosis assay, graph pad prism 9.0.1 (San Diego, CA, USA) was used. Non-linear regression analysis was used to fit the dose-response curves (variable slope model) and calculate absolute IC50 values based on concentration. Statistical analysis was performed using one-way ANOVA. Dunnets posthoc analysis was used to correct multiple comparisons when comparing untreated control with treated samples. To compensate for the violation of homogeneity of variances due to unequal sample sizes, the Welch ANOVA test was applied. The significance of the differences is determined if the *p*-value is less than 0.05.

## Figures and Tables

**Figure 1 molecules-26-07447-f001:**
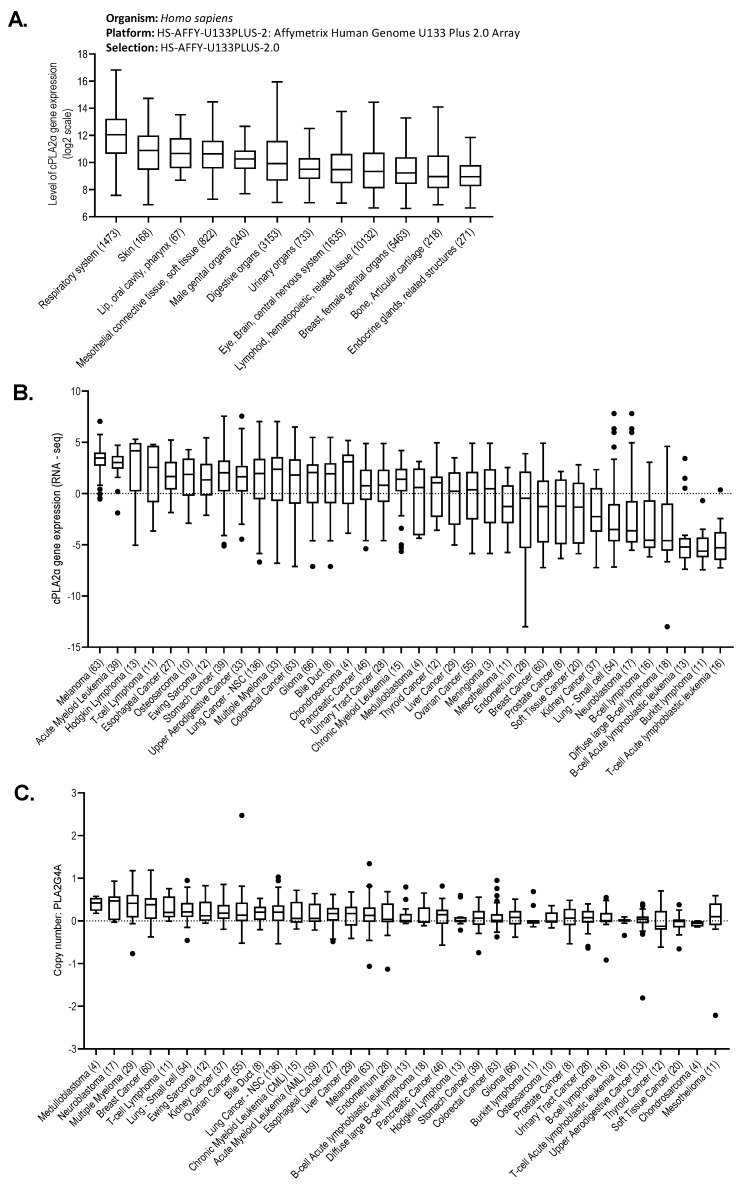
Comparative analysis of *PLA2G4A* gene expression in different cancers and cancer cell lines. (**A**) Relative expression of the *PLA2G4A* gene in 24,375 cancer patient samples grouped by tissue origin (collected from publicly available databases using Genevestigator). (**B**) Relative *PLA2G4A* gene expression (from RNAseq data) in solid and hematological cancer cell lines (collected from CCLE). (**C**) Copy number of the *PLA2G4A* gene in solid and hematological cancer cell lines (collected from CCLE).

**Figure 2 molecules-26-07447-f002:**
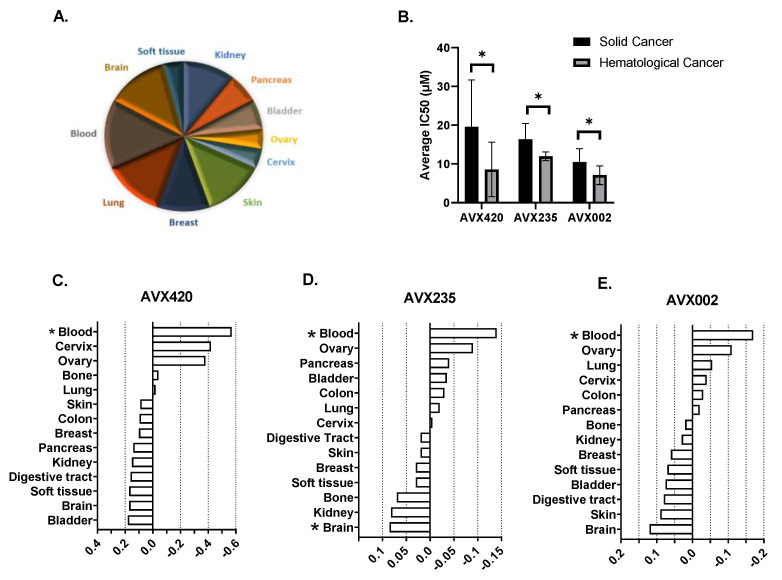
Cellular profiling of cPLA2α inhibitors. (**A**) Tissue sources of the cell lines used in the Oncolines panel. (**B**) Average IC50 values of AVX420, AVX235, and AVX002 in solid versus blood cancer cell lines. To compensate for the violation of homogeneity of variances due to unequal sample sizes, the Welch ANOVA test was applied. * *p* < 0.05. (**C**–**E**): Relative sensitivity of tissue type compared to the panel average is expressed in 10logIC50 for AVX420, AVX235, and AVX002. The chart depicts the average 10logIC50 values for each tissue type, containing at least two cell lines, relative to the average 10logIC50 measured in the entire panel. Tissue types represented by a single cell line (e.g., prostate, thyroid, uterus, etc.) were excluded from the analysis. A number of −1 on the horizontal axis, therefore, reflects a 10x lower IC50 compared to the panel average. The significance of the differences is determined with a t-test and considered significant if the *p*-value is less than 0.05; this is indicated with an asterisk.

**Figure 3 molecules-26-07447-f003:**
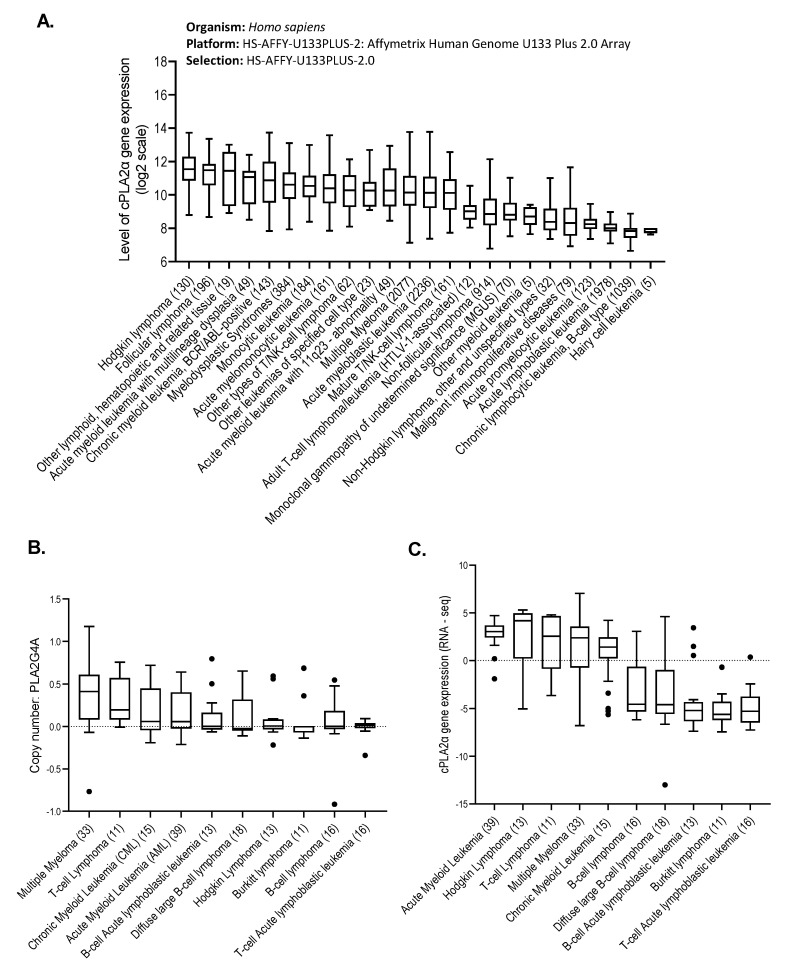
*PLA2G4A* gene expression in hematological cancers. (**A**) *PLA2G4A* gene expression in 10,131 patient samples from more than 20 different lymphoid and hematopoietic cancer types (collected from publicly available databases using Genevestigator). (**B**) Copy number of *PLA2G4A* gene in hematological cancer cell lines (collected from CCLE). (**C**) *PLA2G4A* gene expression (RNA-seq data) in hematological cancer cell lines (collected from CCLE). (**D**) *PLA2G4A* gene expression in 74 multiple myeloma patient samples compared to plasma cells from 37 normal individuals. (**E**) *PLA2G4A* gene expression in 44 monoclonal gammopathy of undetermined significance (MGUS) patient samples compared to bone marrow cells from 22 normal individuals. (**F**) *PLA2G4A* gene expression in three multiple myeloma cell lines (RPMI8226, INA6, and JJN3). Data was obtained from Jonathan Keats’ lab (www.keatslab.org, accessed on 8 February 2021). (**G**) In-house screening of *PLA2G4A* gene expression in three multiple myeloma cell lines (INA6, JJN3, and IH1).

**Figure 4 molecules-26-07447-f004:**
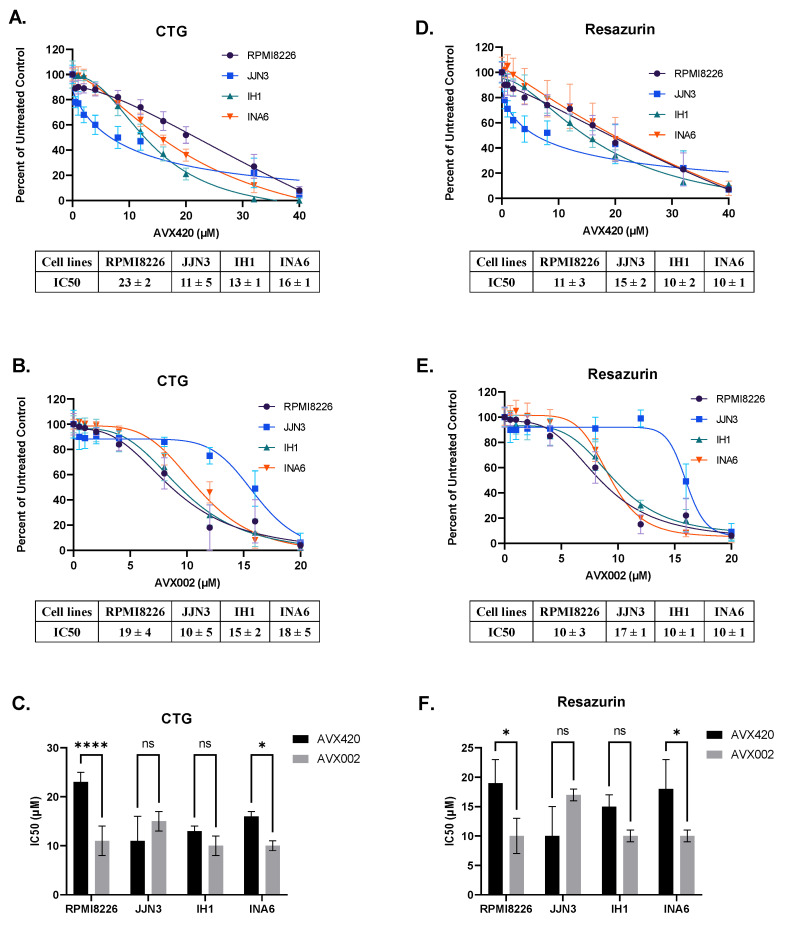
The effect of cPLA2α inhibition on the viability of multiple myeloma cell lines. Cells were treated with the inhibitors (in the range of 0.5 µM to 40 µM) for 72 h before cell titer Glo (CTG) and resazurin viability assays were performed. (**A**,**B**): Dose-response curves for AVX420 and AVX002 were measured using the CTG viability assay. (**C**) Comparison of IC50 values for AVX420 and AVX002 in RPMI8226, JJN3, IH1, and INA6 cells; measured using the CTG viability assay. (**D**,**E**): Dose-response curves for AVX420 and AVX002 were measured using the resazurin viability assay. (**F**) Comparison of IC50 values for AVX420 and AVX002 in RPMI8226, JJN3, IH1, and INA6 cells; measured using the resazurin viability assay. Data are presented as the mean ± standard deviation, and each measurement was repeated at least three times independently. * *p* < 0.05, **** *p* < 0.0001, compared as indicated in the figures.

**Figure 5 molecules-26-07447-f005:**
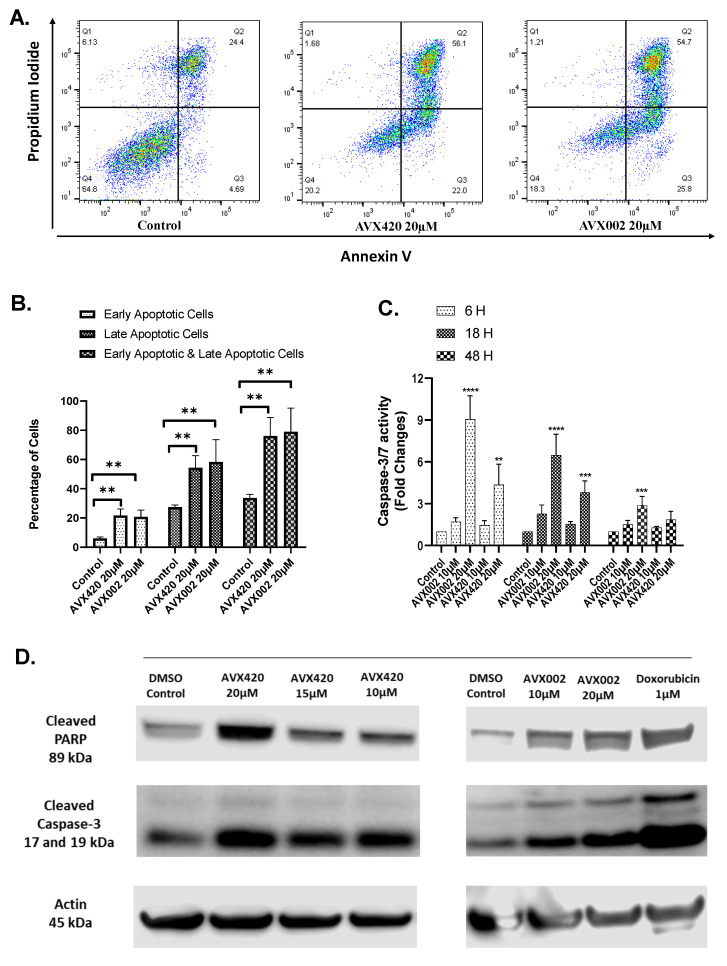
cPLA2α inhibitors induce apoptosis in JJN3 cells. (**A**,**B**): The percentage of early apoptotic, late apoptotic/dead, and a combination of early and late apoptotic/dead cells after 72 h treatment with inhibitors as indicated. (**C**) Measurement of caspase-3/7 activity. Cells were treated with inhibitors as indicated or left untreated (control) for 6 h, 18 h, or 48 h, and caspase-3/7 activity was measured using the caspase-3/7 Glo assay. (**D**) Immunoblot analysis of apoptosis-related proteins after treatment with inhibitors for 36 h. Data in (**B**,**C**) are presented as mean ± standard deviation, and each measurement was repeated at least three times independently. ** *p* < 0.01, *** *p* < 0.001, **** *p* < 0.0001, compared with untreated control.

**Figure 6 molecules-26-07447-f006:**
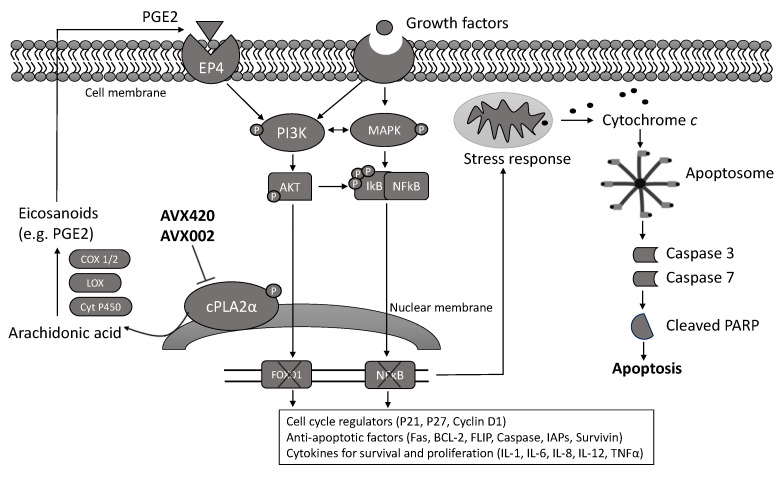
A hypothesized molecular mechanism demonstrates how cPLA2α inhibition could reduce the viability of multiple myeloma cells. Growth factors (e.g., PDGF) are well known to activate MAPK and PI3K/AKT signaling pathways. This can cause the phosphorylation and activation of cPLA2α resulting in the release of arachidonic acid and subsequent production of eicosanoids. PGE2 is one of the most studied eicosanoids, and it is known to bind to and activate the EP4 receptor. EP4 activation can stimulate the PI3K/AKT/FOXO1 and PI3K/AKT/NFκB signaling pathways associated with promoting cell survival and proliferation via regulating the expression of cell cycle proteins, anti-apoptotic factors, and cytokines. Thus, inhibition of cPLA2α by AVX420 and AVX002 may inhibit the pathways that support cell survival and proliferation and cause cellular stress responses leading to the production of reactive oxygen species (ROS) and release of cytochrome *c* from mitochondria, and ultimately the cleavage and activation of caspase-3/7 and PARP to execute apoptosis. PDGF, platelet-derived growth factor; MAPK, mitogen-activated protein kinase; cPLA2α, cytosolic phospholipase A2α; PGE2, prostaglandin E2; COX-1/2, cyclooxygenase-1 and -2; LOX, lipoxygenase; Cyt P450, cytochrome P450; EP4, E-type prostanoid receptor 4; PI3K, phosphatidyl inositol-3 kinase; AKT, protein kinase B; IkB, inhibitor of nuclear factor kappa B; NFκB, nuclear factor kappa B; FOXO1, forkhead box protein O1; PARP, poly (ADP-ribose) polymerase.

## Data Availability

The data that support the findings of this study are available in this manuscript and the databases of Jonathan Keat’s Lab (www.keatslab.org), CCLE (https://sites.broadinstitute.org/ccle/), Genevestigator (https://genevestigator.com/), and Oncomine (https://www.oncomine.org/).

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
