# Peer review of "Inhibition of Cytosolic Phospholipase A2α Induces Apoptosis in Multiple Myeloma Cells"

_molecules, 2021, doi:10.3390/molecules26247447_

Round 1

Reviewer 1 Report

This is an interesting study. The authors first explored public database showing that cPLA2α highly expressed in human cancers of specific tissue origins. Then they found that inhibition of cPLA2α significantly reduced the viability of multiple cell lines. The mechanisms could be the caspase-3 mediated apoptosis. It would be better if the authors use TUNEL assay to directly show the apoptotic cells, but not required in this study.

Overall, this manuscript is well-written, and the approach and conclusion are appropriate. It can be accepted in the present form from my side.

Reviewer 2 Report

In this paper, authors deal with a very important issue,   It is well written with adequate language. The efforts of the authors are appreciated. However there are some modifications should be covered to further improve the paper: authors are advised to add some diagrams illustrate the mechanism of action of the studied enzyme

Author Response

Response to Reviewer 02 comments

Reviewer comment: In this paper, authors deal with a very important issue, it is well written with adequate language. The efforts of the authors are appreciated. However, there are some modifications should be covered to further improve the paper: authors are advised to add some diagrams illustrate the mechanism of action of the studied enzyme

Reply to Reviewer 2: We appreciate the time and efforts that you have dedicated to providing us with your valuable feedback on our manuscript. We agree with your comment. Therefore, we have added figure 6 with text in lines no 318-337 and discussed the figure in lines no 311-318 (Please see the attachment). Thank you for pointing this out.

Reviewer 3 Report

Mohammad et al. present an interesting study in which they investigate the inhibition of cPLA2a in multiple myeloma cells and its effect on induction of cell death. Besides myeloma cells they have included other types of cancer cells. Results are promising for subsequent studies and its application as a therapy against multiple myeloma in the future. The manuscript is well structured and has an exposition logic in which the results can be viewed logically throughout the text. The methods are adequate to answer the research questions, and the selected graphs and figures are adequate. However, I have some comments:

  1. The results of Figure 1 are clear; however, it is necessary to establish what the reference expression level is to mention the level at which cPLA2α is overexpressed in the different tissues.
  2. Figure 2: Why does the word ovary appear twice in 2C? What is the meaning of the different shades of gray in "C, D, and E”?
  3. Results: It is not clear if the 10,022 samples of lymphoid and hematological cancer are from some database or if it was done experimentally for this study. This information and that of the other graphs should also appear in the description of figure 3.
  4. Discussion: I think the authors may add information on what is known about the toxicity of AVX molecules in normal cells or tissues, since it is possible that at some point these molecules could be used in humans.

Author Response

Response to Reviewer 03 comments

Reply to Reviewer: We appreciate the time and efforts that you have dedicated to providing us with your valuable feedback on our manuscript. We agree with your comments and respond accordingly in the edited manuscript file (Please see the attachment).  

Point 1: The results of Figure 1 are clear; however, it is necessary to establish what the reference expression level is to mention the level at which cPLA2α is overexpressed in the different tissues.

Reply to point 1: You have raised an important point here. However, we showed the level of cPLA2a expression among cancers of different tissue origins. Here, cPLA2a was found differentially expressed and we plotted the increasing level of expression from right to left in each figure of figure 1 (line no 111). To our knowledge, no reference expression level of cPLA2a was found.

Point 2: Figure 2: Why does the word ovary appear twice in 2C? What is the meaning of the different shades of gray in "C, D, and E”?

Reply to point 2: It was a typing error in figure 2C (Line no 139). There is no specific meaning of different shades in figure 2C-E (Line no 139). Thank you for pointing this out. We have corrected them.

Point 3: Results: It is not clear if the 10,022 samples of lymphoid and hematological cancer are from some database or if it was done experimentally for this study. This information and that of the other graphs should also appear in the description of figure 3.

Reply to point 3: In the description of figure 3, we have now clearly mentioned about source of those data. See lines no 183, 184, 185. Thank you for your suggestion.

Point 4: Discussion: I think the authors may add information on what is known about the toxicity of AVX molecules in normal cells or tissues, since it is possible that at some point these molecules could be used in humans.

Reply to point 4: We agree with you. We have, accordingly, described the non-toxic effects of the AVX compound from our previous work in the discussion chapter (line no 302-310).

Reviewer 4 Report

This manusucript demonstrated that hematological malignancies are particularly sensitive to the growth inhibitory effect of cPLA2α inhibition, and AVX420 and AVX002 could decrease the viability of multiple myeloma cells and induce apoptosis. This manuscript was well well performed and written. It is acceptable for publication. There are some concerns about the mascript.

  1. Why did the author select 6, 18 48 hour to study AVX420 and AVX002? Why not other time points?
  2. For western blot, line 403, the antibody dilution ratio was not described.
  3. For Figure 4, the author did not interpret what does ***and * mean.

Author Response

Response to Reviewer 04 comments

Reply to Reviewer: We appreciate the time and efforts that you have dedicated to providing us with your valuable feedback on our manuscript. We agree with your comments and respond accordingly in the edited manuscript (Please see the attachment). 

Point 1: Why did the author select 6, 18 48 hour to study AVX420 and AVX002? Why not other time points?

Reply to point 1: It was a time series to accommodate the possible time frame for high caspase 3/7 activity. As the conversion of a pro to active caspase 3/7 is an early phase of apoptosis, we were investigating very early (6H), early (18H), and late (48H) hours to visualize caspase 3/7 activity. We hope that explains the reason for selecting this three-time point.

Point 2: For western blot, line 403, the antibody dilution ratio was not described.

Reply to point 2: We have edited the dilution factors in the text (line no 434). Thank you for pointing this out.

Point 3: For Figure 4, the author did not interpret what does ***and * mean.

Reply to point 3: We have explained the meaning of * in texts (line no 221-222). Thank you for your salient observation.
